# The African ape-like foot of *Ardipithecus ramidus* and its implications for the origin of bipedalism

Thomas Cody Prang[1,2]*

[1]Department of Anthropology, Center for the Study of Human Origins (CSHO), New York University, New York, United states; [2]New York Consortium in Evolutionary Primatology (NYCEP), New York, United States

**Abstract** The ancestral condition from which humans evolved is critical for understanding the adaptive origin of bipedal locomotion. The 4.4 million-year-old hominin partial skeleton attributed to *Ardipithecus ramidus* preserves a foot that purportedly shares morphometric affinities with monkeys, but this interpretation remains controversial. Here I show that the foot of *Ar. ramidus* is most similar to living chimpanzee and gorilla species among a large sample of anthropoid primates. The foot morphology of *Ar. ramidus* suggests that the evolutionary precursor of hominin bipedalism was African ape-like terrestrial quadrupedalism and climbing. The elongation of the midfoot and phalangeal reduction in *Ar. ramidus* relative to the African apes is consistent with hypotheses of increased propulsive capabilities associated with an early form of bipedalism. This study provides evidence that the modern human foot was derived from an ancestral form adapted to terrestrial plantigrade quadrupedalism.
DOI: https://doi.org/10.7554/eLife.44433.001

## Introduction

Terrestrial bipedalism is widely regarded as a shared-derived characteristic of the hominin clade and understanding its evolution is one of the central foci of biological anthropology (*Darwin, 1871*; *Wasburn, 1967*; *Fleagle et al., 1981*; *Richmond et al., 2001*; *Gebo, 1996*; *Begun, 2004*; *Lovejoy et al., 2009a*; *White et al., 2015*). There are numerous adaptive explanations for the origin of bipedalism (*Darwin, 1871*; *Hewes, 1961*; *Lovejoy, 1981*; *Rose, 1991*; *Washburn, 1960*; *Hunt, 1996*) that are inherently difficult to test directly (*Smith and Wood, 2017*), but each of them depends on alternative hypothetical models for the morphology and locomotor behavior of the human-chimpanzee last common ancestor (LCA; *Richmond et al., 2001*). Hypotheses for the locomotor behavior of the LCA include vertical climbing (*Stern, 1975*; *Prost, 1980*; *Fleagle et al., 1981*), terrestrial knuckle-walking (*Wasburn, 1967*; *Gebo, 1992*; *Gebo, 1996*; *Pilbeam, 1996*; *Richmond and Strait, 2000*; *Richmond et al., 2001*; *Begun, 2004*; *Inouye and Shea, 2004*), below-branch suspension (*Keith, 1923*; *Tuttle, 1969*; *Young et al., 2015*), arboreal bipedality (*Thorpe et al., 2007*), and more generalized quadrupedalism with slow, deliberate climbing (*Lovejoy et al., 2009a*; *White et al., 2009*; *White et al., 2015*). These behavioral hypotheses make different assumptions about whether the LCA was adapted to arboreality or terrestriality (*Wasburn, 1967*; *Gebo, 1996*; *Gebo, 1992*; *Schmitt, 2003*; *Crompton et al., 2010*). No matter the specific elements of the proposed behavioral hypothesis for the *Homo-Pan* LCA, hominin bipedalism is either the result of an initial evolutionary shift towards terrestriality from an arboreal ancestor (*Schmitt, 2003*; *Lovejoy et al., 2009a*; *Lovejoy et al., 2009b*; *Lovejoy et al., 2009c*; *White et al., 2009*; *Crompton et al., 2010*; *White et al., 2015*) or, alternatively, a secondary shift from a semi-

*For correspondence:
cody.prang@nyu.edu

**Competing interests:** The author declares that no competing interests exist.

**eLife digest** Walking on two legs is considered to be one of the first steps towards becoming human. While some animals are also able to walk on two legs, such as kangaroos, birds, and some rodents, the way they move is nevertheless quite distinct to the way humans walk.

How animals evolve traits is influenced by the characteristics of their ancestors. But what exactly was the common ancestor of humans and chimpanzees like? Most primates are suited for a life in the trees. But some also have skeletal characteristics associated with living on the ground. For example, the feet of chimpanzees and gorillas show adaptations that suit life on the ground, such as walking on the sole of the foot with a heel first foot posture. So far, it was unclear whether the ancestor of humans and chimpanzees was primarily adapted to living on the ground or in the trees.

To investigate this further, Prang studied the oldest-known fossil foot (4.4 million years) attributed to the hominin *Ardipithecus ramidus.* This involved using evolutionary models to evaluate the relationship between foot bone proportions and the locomotory behaviour of monkeys and apes. The results revealed that humans evolved from an ancestor that had a foot similar to living chimpanzees and gorillas. The African ape foot is uniquely suited to life on the ground, including shorter toe bones, but also shows some adaptations to life in the trees, such as an elongated, grasping big toe. Therefore, the locomotion of our common ancestor probably bore a strong resemblance to these two ape species. Moreover, if the last common ancestor already had ground-living characteristics, the first step of the evolution of human bipedalism did not involve descending from the trees to the ground, as our ancestors had already achieved this milestone in some form and frequency.

This is an important discovery. If this ancestor already had adaptations for life on the ground, why did only humans evolve to walk upright despite the retention of climbing capabilities in the earliest human relatives? A next step could be to investigate what selective pressures favored upright walking in a partly ground-living African ape. This may provide us with more insight into our own evolutionary story as well as the ways in which living primates evolve adaptations in an ecological context.

DOI: https://doi.org/10.7554/eLife.44433.002

terrestrial quadrupedal ancestor (*Figure 1*; *Wasburn, 1967*; *Gebo, 1992*; *Gebo, 1996*; *Pilbeam, 1996*; *Richmond and Strait, 2000*; *Richmond et al., 2001*; *Inouye and Shea, 2004*).

The partial skeleton of the early hominin *Ardipithecus ramidus* purportedly lacks postcranial specializations associated with hominoid-like orthogrady, vertical climbing, and suspension (*Lovejoy et al., 2009a*; *Lovejoy et al., 2009b*; *Lovejoy et al., 2009c*; *White et al., 2009*; *White et al., 2015*), which challenges conventional understandings of ape and human ancestry (*Darwin, 1871*; *Keith, 1923*; *Morton, 1922*; *Wasburn, 1967*; *Gebo, 1992*; *Gebo, 1996*; *Williams, 2012*). In particular, the foot of *Ar. ramidus* was argued to possess monkey-like midfoot stabilizing and propulsive morphologies that were inferred to be primitive for great apes (*Lovejoy et al., 2009a*). Lovejoy and colleagues (*Lovejoy et al., 2009a*; *White et al., 2015*) suggested that the *Ar. ramidus* foot was consistent with 'above branch plantigrady' and a form of locomotion termed 'arboreal multigrady' (*White et al., 2015*). The African apes are viewed as 'adaptive cul-de-sacs' (*Lovejoy et al., 2009b*: pg. 104) that independently evolved adaptations for vertical climbing given the purported lack of 'the peculiar substrate-conforming, hand-like grasping foot of living African apes' (*White et al., 2015*: pg. 4883) in *Ar. ramidus* and in White and colleagues' reconstruction of the *Homo-Pan* LCA. This interpretation of the *Ar. ramidus* foot is consistent with hominin bipedalism emerging from a more generalized, exclusively arboreal, quadrupedal ancestor (*Straus, 1949*; *Schmitt, 2003*; *Lovejoy et al., 2009a*; *Lovejoy et al., 2009b*; *Crompton et al., 2010*).

Foot proportions (e.g., tarsal, metatarsal, and phalangeal lengths) are hypothesized to reflect variation in locomotor behavior among anthropoid primates (*Midlo, 1934*; *Schultz, 1963a*; *Schultz, 1963b*; *Jolly, 1967*; *Strasser, 1992*; *Strasser, 1994*). Furthermore, the modern human foot has been highly modified in response to the biomechanical constraints of terrestrial bipedalism (*Morton, 1922*; *Gebo, 1992*; *Harcourt-Smith and Aiello, 2004*). Foot proportions may therefore provide

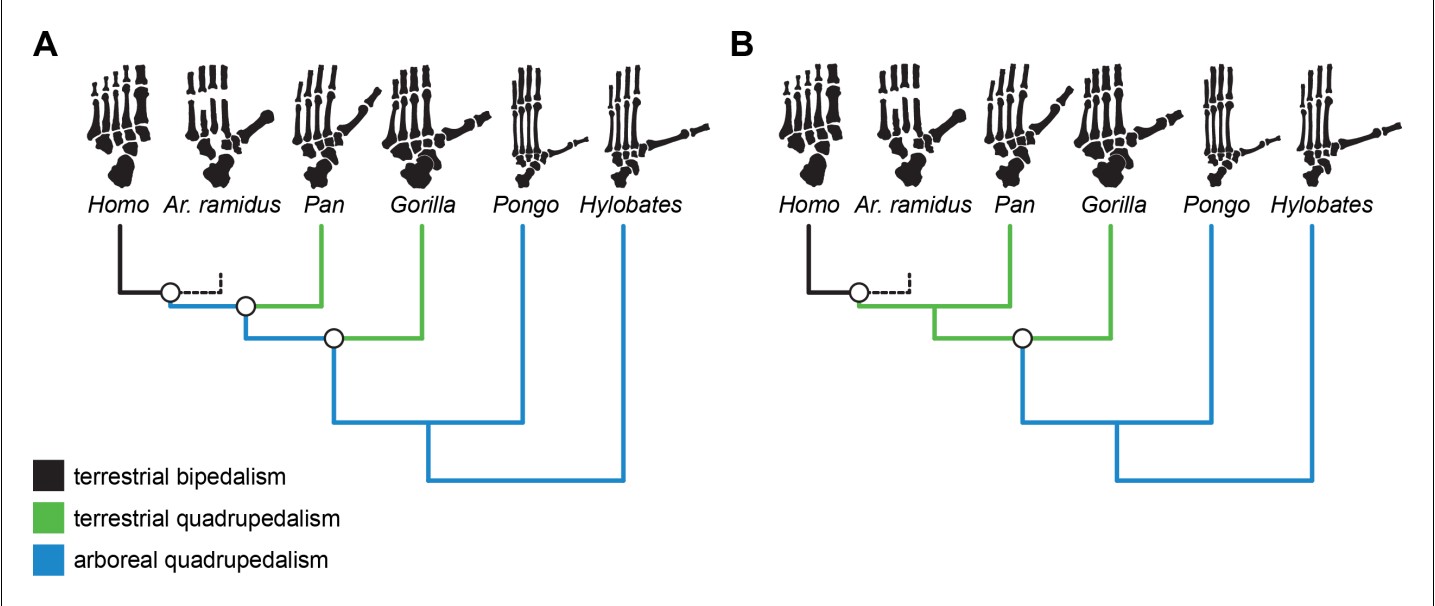

**Figure 1.** Alternative evolutionary scenarios for the origin of hominin bipedalism. Black = terrestrial bipedalism, green = terrestrial quadrupedalism, blue = arboreal quadrupedalism. (**A**) Bipedalism is principally a terrestrial adaptation derived from a more exclusively arboreal ancestor, which is consistent with the original interpretation of the *Ar. ramidus* foot. (**B**) Alternative scenario in which bipedalism originates from an ancestor with terrestrial quadrupedal adaptations, which would be predicted based on the comparative anatomy of living apes and humans. Here, whether a taxon is considered arboreal or terrestrial is based on their reported frequency of arboreality in the wild.

DOI: https://doi.org/10.7554/eLife.44433.003

insight into long-standing debates about alternative models for the *Homo-Pan* LCA (*Keith, 1923*; *Wasburn, 1967*; *Stern, 1975*; *Prost, 1980*; *Fleagle et al., 1981*; *Gebo, 1992*; *Gebo, 1996*; *Richmond et al., 2001*; *Schmitt, 2003*; *Lovejoy et al., 2009b*; *Crompton et al., 2010*; *White et al., 2015*). The locomotor behavior of the earliest hominins significantly alters inferences about the paleobiology of the *Homo-Pan* LCA and our understanding of how bipedalism evolved. This study utilizes recent methodological advances to (1) test alternative hypotheses about the relationship between foot proportions and locomotor behavior among extant anthropoid primates, (2) character-ize the morphometric affinities of the *Ar. ramidus* foot (ARA-VP-6/500) on the basis of foot propor-tions, and (3) estimate the foot proportions of the *Homo-Pan* LCA. Specifically, this study uses a combination of evolutionary modeling (*Hansen, 1997*; *Butler and King, 2004*; *Ingram and Mahler, 2013*) and ancestral state estimation (*Elliot and Mooers, 2014*) methods to make inferences about the evolutionary history of foot proportions in the anthropoid clade and its implications for the adap-tive origin of hominin bipedal locomotion.

## Results

### Morphometric affinities

The morphometric affinities of the *Ar. ramidus* foot were evaluated by constructing a morphospace based on six geometric mean-standardized variables that are preserved in the ARA-VP-6/500 foot skeleton using Principal Components Analysis (PCA, *Figure 2*). The first three principal components account for 96% of the total variance in the sample and clearly separate taxonomic groups along previously hypothesized axes of morphological variation (*Schultz, 1963a*; *Schultz, 1963b*; *Jolly, 1967*; *Strasser, 1992*; *Strasser, 1994*). The first principal component accounts for 63% of the variance and is positively loaded by the lengths of the fifth metatarsal and fourth proximal phalanx. The second principal component accounts for 18% of the variance and is positively loaded by the lengths of the first metatarsal and fourth proximal phalanx, and negatively loaded by the length of

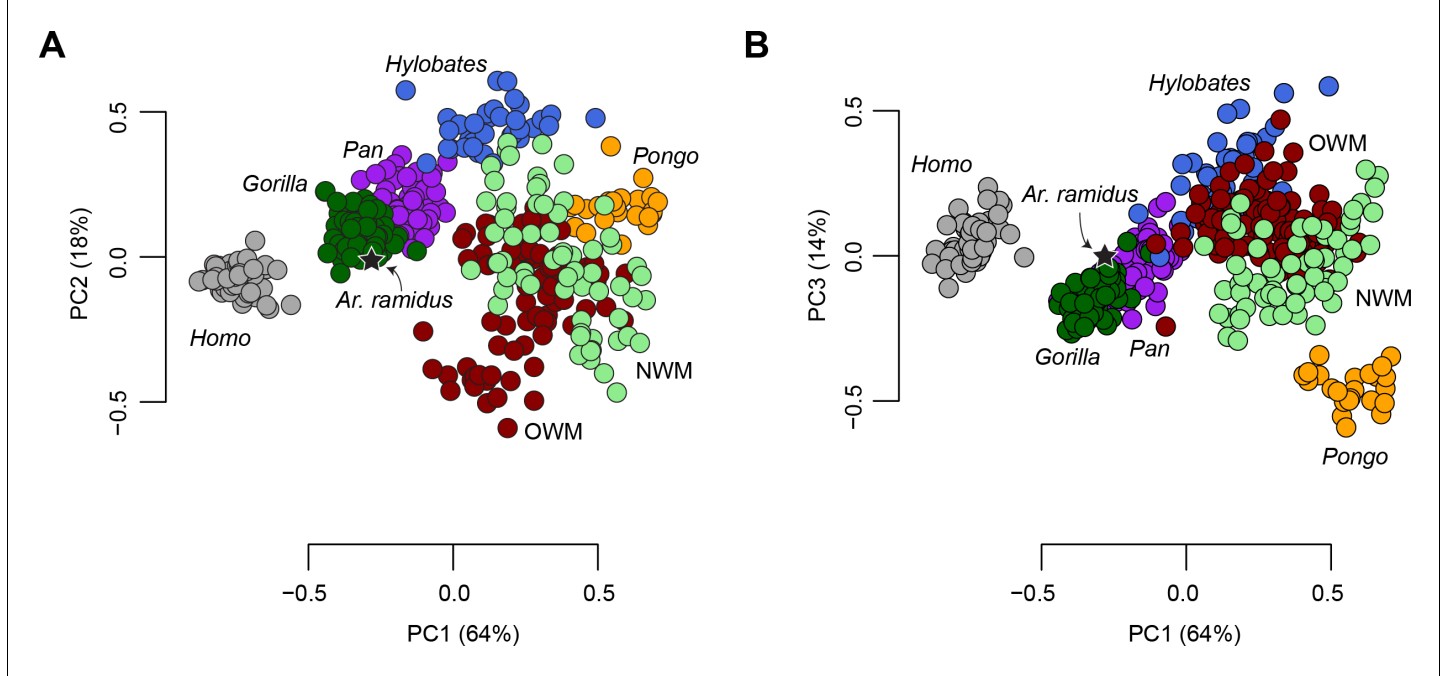

**Figure 2.** Principal Components Analysis (PCA) on six geometric mean-standardized variables. (**A**) The first two principal components representing 82% of the variance. (**B**) The first and third principal components representing 76% of the variance. Star = *Ar. ramidus*, Grey = *Homo*, green = *Gorilla*, purple = *Pan*, orange = *Pongo*, blue = *Hylobates*, red = Old World monkeys, light green = New World monkeys. Note the placement of *Ar. ramidus* with the African apes.

DOI: https://doi.org/10.7554/eLife.44433.004

The following figure supplements are available for figure 2:

**Figure supplement 1.** UPGMA dendrogram on Euclidean distances in anthropoid primates and *Ardipithecus ramidus*.
DOI: https://doi.org/10.7554/eLife.44433.005

**Figure supplement 2.** Univariate comparisons of *Ar. ramidus* foot proportions to extant anthropoids.
DOI: https://doi.org/10.7554/eLife.44433.006

the fifth metatarsal. The third principal component, which represents 15% of the variance, is positively loaded by the length of the first metatarsal (*Table 1*).

The distribution of anthropoid taxa in the PCA is consistent with predictions based on locomotor behavior. For example, the more terrestrial taxa fall at the negative end (*Homo*, *Pan*, *Gorilla*, *Theropithecus*, *Papio*, and *Erythrocebus*) with shorter metatarsals and phalanges, whereas the most arboreal, suspensory, taxa fall at the positive end of PC1 (e.g., *Pongo* and *Ateles*). The terrestrial taxa are

**Table 1.** Principal Components Analysis (PCA).

|  | PC1 | PC2 | PC3 |
|---|---|---|---|
| Eigenvalue | 0.17 | 0.05 | 0.04 |
| Percent variance | 64.6 | 17.8 | 14.1 |
| MT1 length | −0.01 | 0.34 | 0.91 |
| MT5 length | 0.64 | −0.67 | 0.30 |
| PP4 length | 0.68 | 0.63 | −0.18 |
| Talar trochlea length | −0.29 | 0.04 | −0.06 |
| Talar neck length | −0.08 | −0.07 | −0.21 |
| Cuboid length | −0.19 | −0.17 | 0.06 |

DOI: https://doi.org/10.7554/eLife.44433.007

sort into those that are heel-strike plantigrade (*Homo*, *Pan*, and *Gorilla*) and those that are digitigrade (*Theropithecus*, *Papio*, *Erythrocebus*). Hylobatids, atelids, and *Pongo* are distinguished from other arboreal taxa along the same axis that separates terrestrial heel-strike plantigrade taxa from terrestrial digitigrade taxa. A UPGMA cluster analysis (cophenetic correlation coefficient = 0.82) shows that of the 44 extant taxa presented here *Ar. ramidus* is most similar to *Pan* and *Gorilla* (*Figure 2—figure supplement 1*). Univariate comparisons show that *Ar. ramidus* possesses a cuboid that is only slightly elongated relative to African apes (*Figure 2—figure supplement 2A*), a relatively short fourth proximal phalanx (*Figure 2—figure supplement 2B*), and an intrinsically elongated first metatarsal like African apes and atelids (*Figure 2—figure supplement 2C*).

The scaling of various tarsals, metatarsals, and phalanges with body mass was investigated using phylogenetic generalized least squares regression (*p*GLS) to account for the statistical non-independence of the data due to phylogenetic relationships. The parameters for each of the *p*GLS models include the intercepts and slopes of the variables regressed on log body mass, their standard error (s.e.), *T*, and the *p*-value (*Supplementary file 1*). All variables scale with slight negative allometry in that larger species tend to have relatively shorter metatarsals, phalanges, and tarsals. The only exception is the length of the talar trochlea, which scales isometrically with body mass. Pagel's lambda (λ) is a parameter commonly estimated in *p*GLS regression analyses as a measure of phylogenetic signal that can be used to transform the branches of the phylogenetic tree to improve model fit (*Pagel, 1999*; *Revell, 2010*). A λ value of 1 would be consistent with expectations under a Brownian motion evolutionary model. The tarsal measurements show departure from Brownian motion (λ = 0.574 or less), whereas the higher λ values of the metatarsal and phalangeal variables are consistent with a Brownian motion model. Several of the λ values are not significantly different from either 0 or 1, which suggests the true λ for these models is uncertain. The PC scores used in the evolutionary analyses are not correlated with body mass (*p*GLS *p* = 0.08 or higher), which suggests that body mass is not responsible for driving the differences in intrinsic foot proportions among anthropoid groups.

## Evolutionary modeling

The adaptive implications of an African ape-like morphology in *Ar. ramidus* (*Figure 2*) were evaluated using evolutionary modeling. The input data include the first three principal components from the PCA described above. This approach was chosen to reduce the dimensionality of the dataset for evolutionary modeling and ancestral state estimation. Alternative *a priori* evolutionary hypotheses were constructed and include a Brownian motion, single-optimum Ornstein-Uhlenbeck (OU), and several multi-optima OU models (*Butler and King, 2004*) (*Figure 3—figure supplement 1*). Alternative multi-optima OU models were constructed using different selective regimes associated with locomotion in extant taxa. The evolutionary models differ in increasing complexity where each model includes additional phenotypic optima in a hierarchical manner. Model comparisons using multiple criteria (AIC, AICc, SIC) show that the multi-optima OU models are a better fit to the data than a Brownian or single-optimum OU model (*Supplementary file 2*), which suggests there are multiple adaptive peaks associated with foot proportions among anthropoid primates. An additional evolutionary model was constructed without identifying selective regimes *a priori* and resulted in a similar pattern of selective regimes compared to the best fitting *a priori* hypothesis (*Figure 3*). The evolutionary hypothesis with the most favorable AICc value is the most complex and includes selective regimes associated with bipedalism, terrestrial plantigrady, terrestrial semiplantigrady, arboreal quadrupedalism, arboreal quadrupedalism with increased frequency of hindlimb-assisted suspension, and arboreal quadrupedalism with increased frequency of climbing. Simulations show that there is adequate power to distinguish between alternative models and provides support for the *a priori* model selection results obtained using AICc (*Figure 3—figure supplements 1–2*).

## LCA reconstructions

A molecular consensus phylogeny with branch lengths proportional to elapsed time was superimposed on the multivariate data and ancestral values were estimated using a Markov chain Monte Carlo (MCMC) method. This approach relaxes assumptions of neutrality and gradualism and therefore minimizes the effect of exceptional lineage divergences on the estimation of ancestral values. The estimated value for the *Homo-Pan* LCA is nearest to the African apes and highly distinct from all

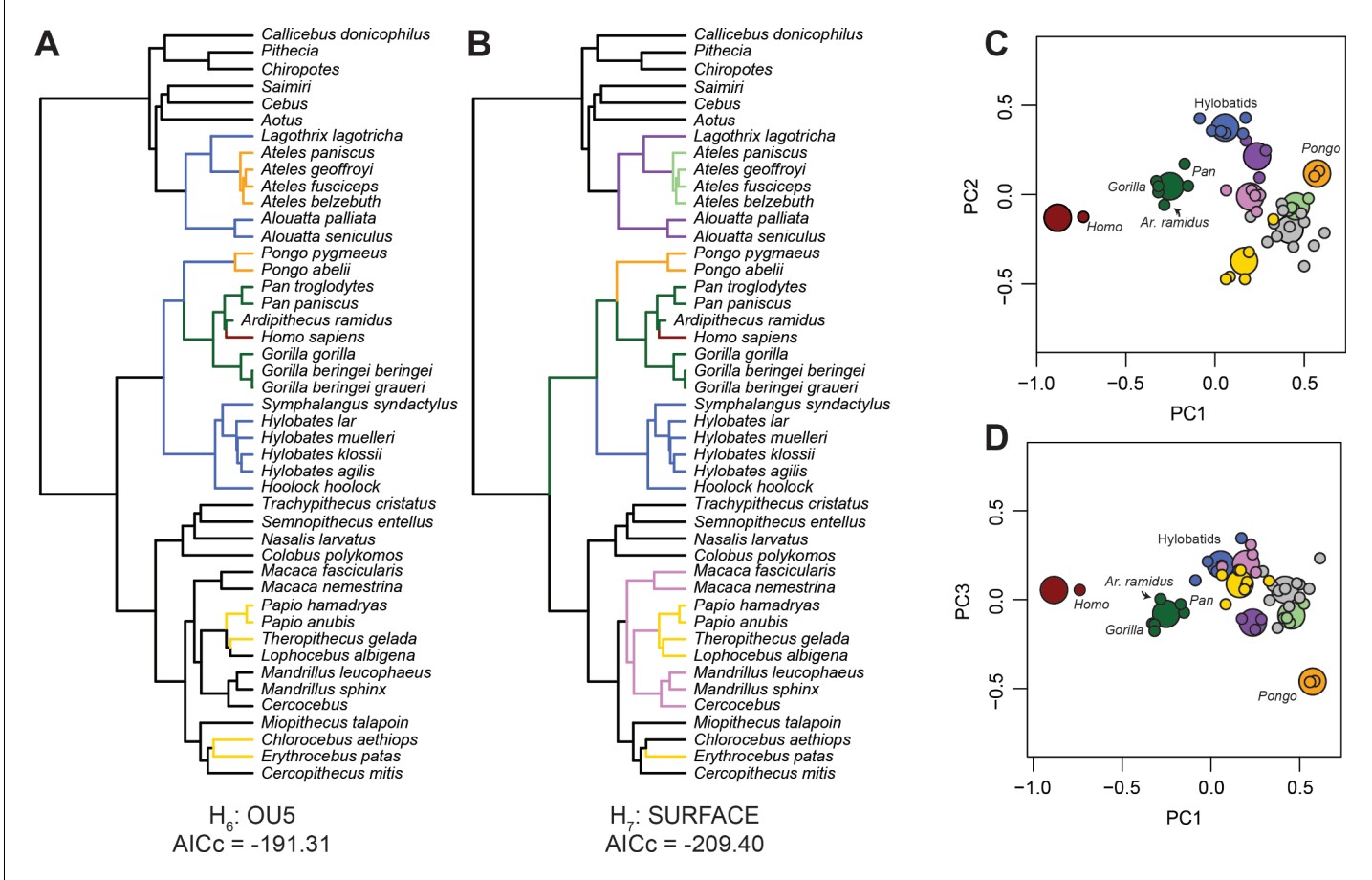

**Figure 3.** Evolutionary modeling. Best fitting evolutionary models. (**A**) Best fitting a priori evolutionary hypothesis according to OUCH. (**B**) Arrangement of selective regimes fit by SURFACE. (**C**) The first two principal components with phenotypic optima estimated by SURFACE. (**D**) The first and third principal components with phenotypic optima estimated by SURFACE. Note the tight fit of species means (small dots) around their optima (large dots) as well as the placement of *Ar. ramidus* near the African ape phenotypic optimum. The colors in C and D correspond to the selective regimes painted onto the phylogeny in B.

DOI: https://doi.org/10.7554/eLife.44433.008

The following figure supplements are available for figure 3:

**Figure supplement 1.** Alternative evolutionary hypotheses represented by painting branches of the phylogeny according to selective regimes.
DOI: https://doi.org/10.7554/eLife.44433.009
**Figure supplement 2.** Simulation results for *a priori* evolutionary model comparisons.
DOI: https://doi.org/10.7554/eLife.44433.010

other taxa (*Figure 4*). The 95% credibility intervals for the PC scores of the node representing the *Homo-Pan* LCA are relatively narrow and include the mean values for *Pan paniscus* and *Gorilla gorilla*. The estimated ancestral values for both hominids and hominoids are nearest to *Alouatta* and *Lagothrix*, which is consistent with prior suggestions based on the comparative morphology of the foot in extant and fossil taxa (*Gebo, 1996*; *Sarmiento, 1983*; *Langdon, 1985*; *Harrison, 1986*).

## Discussion

The multivariate evolutionary modeling analyses presented here confirms that among anthropoid primates intrinsic foot proportions are linked to locomotor behavior (*Midlo, 1934*; *Schultz, 1963a*; *Schultz, 1963b*; *Jolly, 1967*; *Strasser, 1992*; *Strasser, 1994*). The combination of evolutionary modeling with ancestral estimations provides evidence for homoplasy in the evolution of anthropoid foot proportions, which strengthens hypotheses about the link between morphology and behavior. For example, *Erythrocebus* probably evolved a terrestrially adapted, digitigrade foot characterized

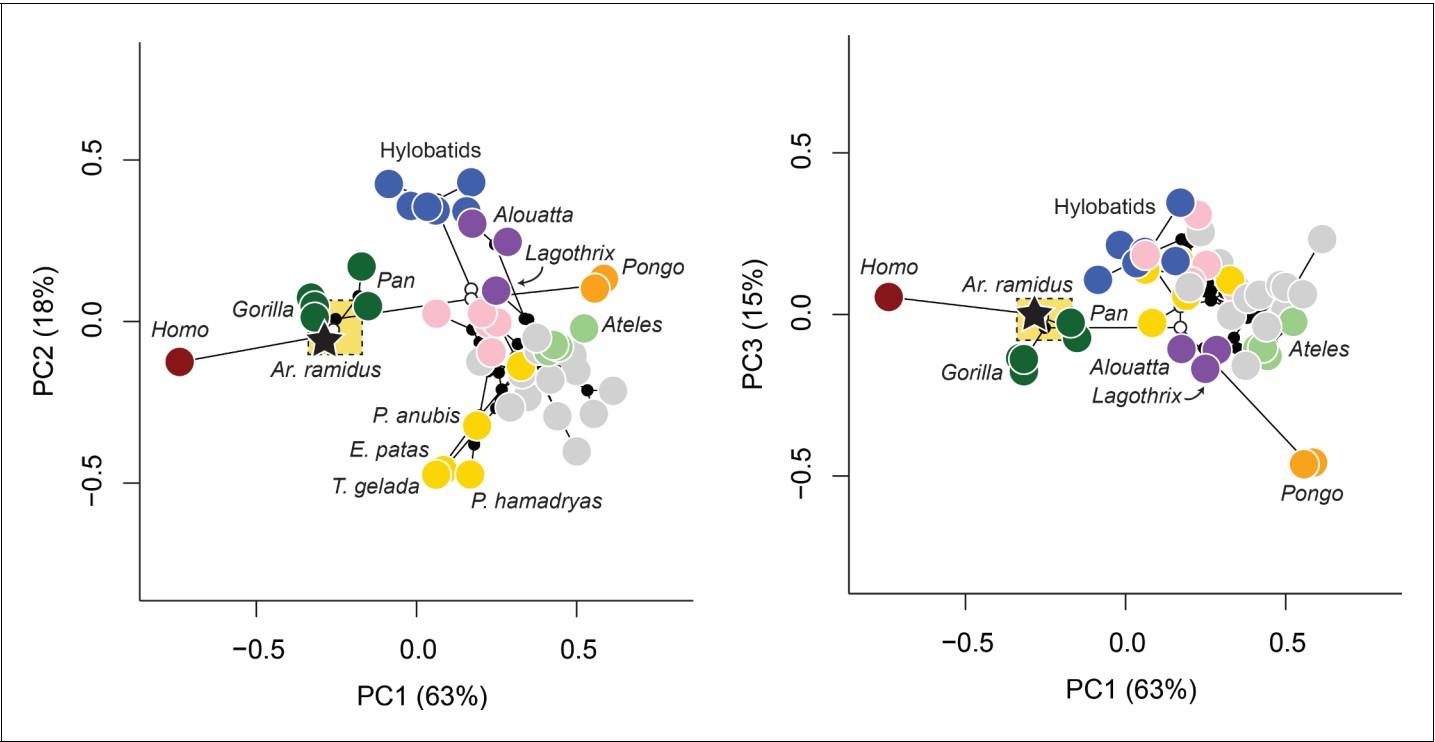

**Figure 4.** Phylomorphospace. Phylomorphospace plots describing the evolution of the anthropoid foot. Internal nodes are represented by black dots with exception for the nodes corresponding to the ancestral values for hominins, hominids, and hominoids which are white. The stitched gold box refers to the 95% credibility intervals for the *Homo-Pan* ancestral state. Note the placement of *Ar. ramidus* with the African apes as well as the separation of taxa according to known differences in locomotion (e.g., more terrestrial taxa are represented by lower values of PC1 whereas more arboreal taxa are represented by higher values of PC1, taxa that engage in active climbing are represented by higher values of PC2). The estimated ancestral morphology for hominins is African ape-like and for both hominids and hominoids it is nearest to *Alouatta* and *Lagothrix*.
DOI: https://doi.org/10.7554/eLife.44433.011

by short phalanges, a short hallux, and long non-hallucal metatarsals independently of *Theropithecus* and *Papio*. These terrestrial monkeys also tend to have a longer midfoot, which increases the pedal load arm and enhances propulsive capabilities. Although most anthropoids sampled here are highly arboreal, the foot of *Alouatta*, *Lagothrix*, and hylobatids converge on similar foot proportions associated with arboreal climbing (though the upper limb of hylobatids is highly autapomorphic), such as a long hallux, relatively shorter metatarsals, and longer phalanges derived from a more generalized anthropoid ancestral condition. The more suspensory anthropoids *Ateles* and *Pongo* may have independently evolved towards the part of the morphospace associated with pedal elongation.

African apes and *Ar. ramidus* occupy a distinct phenotypic optimum characterized by short phalanges, short metatarsals, moderately elongated tarsals, and a long hallux. Although the more terrestrial hominines (*Homo*, *Ardipithecus*, *Pan*, *Gorilla*) and cercopithecines (*Papio*, *Theropithecus*, *Erythrocebus*) probably evolved short phalanges in parallel (*Figures 1* and *2*), the hominines retain an intrinsically elongated hallux and short non-hallucal metatarsals while the terrestrial cercopithecines display the opposite configuration (*Schultz, 1963a*; *Schultz, 1963b*). The difference in foot proportions between terrestrially adapted apes and monkeys are similar to the differences between plantigrade and digitigrade carnivorans (*Taylor, 1976*). The long metatarsals of digitigrade quadrupeds increase the range of plantarflexion at the talocrural joint at the expense of decreasing the effective mechanical advantage of the ankle plantar-flexors (*Biewener, 1989*). African apes and *Ar. ramidus* are separated from terrestrial monkeys along the same axis that distinguishes generalized arboreal anthropoids (e.g., *Cercopithecus*) from hallux-elongated taxa associated with behaviors variously described as 'climbing' (Hylobatids, *Alouatta*, *Lagothrix*). The unique combination of these traits in the hominine foot supports the hypothesis that such feet are adapted to both terrestrial heel-strike plantigrade (rather than digitigrade) quadrupedalism and vertical climbing, which is

consistent with previous suggestions based on comparative anatomy and behavioral observations (*Fleagle et al., 1981*; *Gebo, 1996*; *Prost, 1980*; *Gebo, 1992*; *DeSilva, 2009*; *Prang, 2015*). If *Ar. ramidus* and the *Homo-Pan* LCA were adapted to more generalized quadrupedalism and climbing as originally suggested (*Lovejoy et al., 2009a*; *White et al., 2009*) or 'arboreal multigrady' as later revised (*White et al., 2015*); see also *Fernández et al., 2018*), then the *Ar. ramidus* foot skeleton should not have been placed in the same selective regime as the African apes (*Figure 3*).

The ancestral estimations provide support for the hypothesis that modern humans evolved from an ancestor with African ape-like foot proportions. Modern humans have the longest cuboids of any of the anthropoid primates sampled here (*Figure 2*), which reflects a biomechanical strategy for lengthening the foot's load arm to enhance aspects of propulsion (i.e., range of motion) while simultaneously restricting metatarsal length to minimize bending moments associated with plantigrady (*Lovejoy et al., 2009a*). Although the cuboid elongation of *Ar. ramidus* from the estimated LCA is more modest than previously reported (*Lovejoy et al., 2009a*), it is more parsimoniously interpreted to be derived in the direction of modern humans from an African ape-like ancestor with a short midfoot, rather than an ancestral retention from a more monkey-like great ape ancestor (*Lovejoy et al., 2009a*; *Lovejoy et al., 2009b*; *White et al., 2015*; *McNutt et al., 2018*). Midfoot elongation is consistent with the functional hypothesis of increased propulsive capabilities associated with the form of bipedalism practiced by *Ar. ramidus* (*Lovejoy et al., 2009a*; *White et al., 2015*; *Suwa et al., 2009*; *Kimbel et al., 2014*; *Kozma et al., 2018*; *Fernández et al., 2018*). These analyses show that the origin of bipedalism cannot be explained as an initial shift toward terrestriality from a more exclusively arboreal ancestor (*Figure 3*). Instead, early hominins, including *Ar. ramidus*, evolved from an ancestor with an African ape-like foot adapted to terrestrial plantigrade quadrupedalism (*Morton, 1922*; *Wasburn, 1967*; *Gebo, 1992*; *Gebo, 1996*).

Evidence for terrestrial plantigrade quadrupedalism in the *Homo-Pan* LCA raises the question of whether or not purported knuckle-walking traits in hominines might be homologies (*Richmond et al., 2001*; *Gebo, 1996*; *Begun, 2004*; *Inouye and Shea, 2004*) rather than homoplasies (*Dainton and Macho, 1999*; *Dainton, 2001*; *White et al., 2015*; *Lovejoy et al., 2009b*; *Kivell and Schmitt, 2009*). The knuckle-walking hand posture of African apes is hypothesized to be a secondary adaptation to terrestriality in taxa that retain long hands in association with below-branch forelimb suspension (*Tuttle, 1969*). However, there is no special relationship between plantigrade foot postures and knuckle-walking hand postures (e.g., plantigrade non-primate mammals do not have a knuckle-walking hand posture). Whether or not modern humans evolved from a knuckle-walking ancestor relies on the analysis of the hand and wrist (*Tuttle, 1969*; *Gebo, 1996*; *Dainton, 2001*; *Begun, 2004*; *Inouye and Shea, 2004*; *Dainton and Macho, 1999*; *Kivell and Schmitt, 2009*; *Almécija et al., 2015*). The hand of *Ar. ramidus* was argued not to display traits associated with forelimb suspension in extant taxa or show evidence of a knuckle-walking ancestry (*White et al., 2015*; *Lovejoy et al., 2009b*; *Almécija et al., 2015*), but many of those observations have not yet been independently validated. The terrestrial specializations of the hominine foot are likely to be homologous because they are present in *Ar. ramidus* and they are consistent with model-based ancestral estimations. Critically, they carry a similar set of implications for the origin of bipedalism regardless of the terrestrial hand posture of the *Homo-Pan* LCA.

This study provides evidence that modern humans evolved from an ancestor with an African ape-like foot associated with terrestrial plantigrady and vertical climbing. Hominin upright walking therefore likely emerged in the context of semi-terrestrial quadrupedalism. Explaining the adaptive origin of hominin bipedalism (i.e., 'why' bipedalism evolved) will continue to be a challenging endeavor (*Smith and Wood, 2017*). However, the comparative and fossil material provides evidence for patterns of evolution (i.e., 'how' bipedalism evolved) and strongly suggests that hypotheses of a non-African ape-like morphology for the foot of the *Homo-Pan* LCA (*Lovejoy et al., 2009a*; *White et al., 2015*; *Lovejoy et al., 2009b*) are inconsistent with the results from this study. The *Ar. ramidus* ARA-VP-6/500 partial skeleton is remarkable for its preservation of multiple areas of anatomy (*Lovejoy et al., 2009a*; *White et al., 2009*; *White et al., 2015*). This study provides evidence that intrinsic foot proportions reflect locomotor diversity among anthropoid primates, but it will be important to consider other regions in future comparative studies. The hypothesis that hominins evolved from a semi-terrestrial quadrupedal ancestor could be tested with detailed quantitative analyses of other aspects of *Ar. ramidus* postcranial morphology.

## Materials and methods

### Extant and fossil sample

The extant sample is composed 385 individuals representing 45 taxa: *Homo sapiens*, *Ardipithecus ramidus*, *Pan troglodytes*, *Pan paniscus*, *Gorilla beringei beringei*, *Gorilla beringei graueri*, *Gorilla gorilla*, *Pongo pygmaeus*, *Pongo abelii*, *Hylobates lar*, *Hylobates muelleri*, *Hylobates klossii*, *Hylobates agilis*, *Hoolock hoolock*, *Symphalangus syndactylus*, *Macaca fascicularis*, *Macaca nemestrina*, *Papio anubis*, *Papio hamadryas*, *Theropithecus gelada*, *Lophocebus albigena*, *Mandrillus sphinx*, *Mandrillus leucophaeus*, *Cercocebus* spp., *Erythrocebus patas*, *Chlorocebus aethiops*, *Cercopithecus mitis*, *Miopithecus talapoin*, *Nasalis larvatus*, *Colobus polykomos*, *Trachypithecus cristatus*, *Semnopithecus entellus*, *Ateles geoffroyi*, *Ateles fusciceps*, *Ateles paniscus*, *Ateles belzebuth*, *Alouatta palliata*, *Alouatta seniculus*, *Lagothrix lagotricha*, *Cebus* spp., *Saimiri* spp., *Aotus* spp., *Pithecia* spp., *Chiropotes* spp., and *Callicebus donicophilus*. These specimens are housed at the following collections: American Museum of Natural History (AMNH), Cleveland Museum of Natural History (CMNH), Harvard Museum of Comparative Zoology (MCZ), United States National Museum of National History, Smithsonian Institution (USNM), Field Museum (FM), Berkeley Museum of Vertebrate Zoology (MVZ), Human Evolution Research Center (HERC) at the University of California, Berkeley, and the Royal Museum for Central Africa (RMCA). The modern human sample is composed of recent modern individuals of European and African ancestry housed at the Hamann-Todd Collection at the CMNH as well as a Native American population from California housed at the Phoebe A. Hearst Museum of Anthropology (PAHMA) at the University of California, Berkeley. Measurements of *Ardipithecus ramidus* (ARA-VP-6/500) were initially taken on casts at the University of California, Berkeley. Observations were then made on the original fossils at the National Museum of Ethiopia (NME) and measurements confirmed by T. White.

### Data acquisition

Six measurements were taken on the foot of each individual using Mitutoyo digital calipers: maximum talar articular length, talar trochlea length, cuboid length, first metatarsal (MT1) length, fifth metatarsal length (MT5), and fourth proximal phalanx (PP4) length. Talar neck length was derived by subtracting the talar trochlea length from maximum talar articular length. Maximum talar articular length is defined as the maximum proximodistal distance between the most proximal margin of the talar trochlea and the most distal point on the talar head. Talar trochlea length is defined as the maximum proximodistal distance between the most proximal margin of the talar trochlea and the most distal point of the talar trochlea. Cuboid length is defined as the proximodistal distance between the dorsal margin of the calcaneal facet and the most distal point of the tarsometatarsal joint, taken in dorsal view in approximate anatomical orientation. The purpose of measuring cuboid length in this manner is to explicitly avoid the cuboid beak or calcaneal process since it varies extensively among great apes (*Lewis, 1983*) and because it confounds the cuboid length measurement as a representation of midfoot length since it is articular and housed within a corresponding concavity on the cuboid facet of the calcaneus. MT1 length is defined as the maximum proximodistal distance between the most proximal points on the metatarsal base (with calipers held flush) and the most distal point of the metatarsal head. MT5 length is defined as the proximodistal distance between the most proximal point of the cuboid-MT4 articular margin and the most distal point on the metatarsal head. PP4 length is defined as maximum proximodistal distance between the most proximal point of the phalangeal base and the most distal point of the trochlea. The MT5 was chosen to represent non-hallucal metatarsal length because it is preserved in the ARA-VP-6/500 foot. The fossil is missing most of its metatarsal head and its length was estimated by *Lovejoy et al. (2009a)* using a combination of anatomical and statistical estimation. There is a nearly complete third metatarsal of *Ar. ramidus*, but it derives from a different locality and is therefore associated with a different individual. There is a partially preserved second metatarsal of *Ar. ramidus* also from a different individual (*Lovejoy et al., 2009a*). Therefore, for this study, the metrics are based on the preserved elements of the ARA-VP-6/500 foot of *Ar. ramidus*.

The individual elements of the bony foot skeleton contribute to the production of three movements used in various forms of primate locomotor behavior that are hypothesized to be reflected in intrinsic foot proportions: hallucal adduction and flexion, non-hallucal digital flexion, and

plantarflexion at the talocrural joint. Increasing the length of the first metatarsal should increase the moment arm of the intrinsic adductor musculature such as the m. adductor hallucis across a range of hallucal abduction angles, and therefore should increase the hallucal adduction force during grasping in non-human primates (*Cartmill, 1979*). Increasing hallucal metatarsal and non-hallucal phalangeal lengths also contributes to increasing the span of the pedal grasp in taxa with a mobile hallux, which also helps to maintain a friction grip in pedal grasping (*Cartmill, 1979*). Previous studies have modeled the foot skeleton as a second-class lever, where the fulcrum is at the metatarsophalangeal joints, the load passes through the talocrural joint at the rearfoot, and the force is produced by the plantarflexor muscles. The load arm is the distance between the fulcrum and the load, whereas the effort arm (or 'power arm') is the distance between the insertion of the ankle plantarflexors on the calcaneal tuberosity and the talocrural joint (*Schultz, 1963a*; *Schultz, 1963b*; *Strasser, 1992*). Increasing the effort arm of the foot relative to the load arm increases the mechanical advantage of the foot skeleton as a lever (*Schultz, 1963a*; *Schultz, 1963b*; *Strasser, 1992*). However, increasing the load arm increases the range of motion for a given amount of plantarflexor contraction (*Schultz, 1963a*; *Schultz, 1963b*). The length of the effort arm of the *Ar. ramidus* foot is unknown because its calcaneus is not well preserved (*Lovejoy et al., 2009a*). There are multiple anatomical strategies for increasing the length of the foot skeleton's load arm. The length of the load arm can be increased by lengthening the metatarsals, tarsals (e.g., the talus and/or cuboid), or any combination of these elements. Increasing the length of the metatarsals subjects them to greater bending moments during stance phase, and therefore increases the possibility of injury, so humans achieve a longer load arm by instead increasing the length of the tarsals. As such, in a bipedal heel-strike plantigrade foot, the load arm is increased by lengthening the cuboid and other midtarsal elements. In contrast, in quadrupedal semiplantigrade or digitigrade primates, and indeed other terrestrial cursorial mammals (*Taylor, 1976*), the load arm is lengthened by increasing the length of the metatarsals. One implication of these differing anatomical arrangements (e.g., increasing metatarsal versus tarsal lengths), is that foot proportions may also be a correlate of foot postures (plantigrady and digitigrady or semiplantigrady), which is one of the hypotheses tested by this study using evolutionary modeling.

## Statistical analysis

To correct for differences in scale among species, each measurement was divided by the geometric mean of all six measurements per individual (*Jungers et al., 1995*). Morphometric affinities were evaluated using an unweighted pair-group with arithmetic mean (UPGMA) cluster analysis on Euclidean distances. The cophenetic correlation coefficient was used to assess the degree to which the resulting dendrogram represented the true pairwise distances between taxa (*Sokal and Rohlf, 1962*). Principal Components Analysis (PCA) on all six geometric mean-standardized variables was used to reduce, ordinate, and visualize the multivariate data. All evolutionary modeling and ancestral state estimation analyses use the first three principal components (PCs) derived from the PCA. The PC scores were used instead of the original data to avoid analytical problems surrounding correlations among variables (*Clavel et al., 2015*) and to maximize statistical power (*Boettiger et al., 2012*). *Ardipithecus ramidus* was added to a molecular phylogenetic tree from 10 k trees (*Arnold et al., 2010*) as a stem hominin (*Strait and Grine, 2004*; *White et al., 2009*; *Dembo et al., 2015*) with a branch length of 1.4 million years in accordance with first and last appearance data for the genus (5.8–4.4 Ma (*Haile-Selassie, 2001*; *WoldeGabriel et al., 2001*)) using Mesquite software (*Maddison and Maddison, 2017*). The branch length for *Homo sapiens* was reduced in order to improve estimation of phenotypic optima in evolutionary analyses (*Butler and King, 2004*; *Ingram and Mahler, 2013*). The scaling of individual foot elements with body mass among extant taxa (*Smith and Jungers, 1997*) was conducted using phylogenetic generalized least squares regression (*Grafen, 1989*) with the 'caper' package (*Orme, 2013*) in R (*R Core Team, 2017*). To increase the fit of the evolutionary model to the data, branch lengths were transformed using Pagel's lambda (*Pagel, 1999*), which was estimated with maximum likelihood as a measure of phylogenetic signal in the residual error of each $p$GLS model (*Revell, 2010*).

A phylomorphospace was constructed by superimposing the phylogenetic tree on the average principal component (PC) scores for each taxon and ancestral values were estimated using a Markov chain Monte Carlo method (MCMC) that relaxes assumptions of neutrality and gradualism (*Elliot and Mooers, 2014*). Ancestral values estimated under constant rate Brownian motion are

affected by taxa that are exceptionally phenotypically derived compared to their close relatives because it is assumed that all branches have evolved at the same rate (*Elliot and Mooers, 2014*). The assumption of a constant evolutionary rate therefore results in an 'averaging effect' of ancestral values (*Schluter et al., 1997*; *Elliot and Mooers, 2014*). Therefore, estimates of ancestral values for continuous traits that assume a constant evolutionary rate are potentially biased in the direction of more derived branches characterized by higher phenotypic evolutionary rates. Since this study is focused on estimating ancestral values for humans, great apes, and hominoids, the stable model (*Elliot and Mooers, 2014*) was specifically chosen in light of evidence for molecular and morphological evolutionary rate differences in hominoids relative to other anthropoids (*Steiper et al., 2004*) and in *Homo* relative to *Pan* (*Weaver and Stringer, 2015*).

Ancestral states (PC scores) were estimated under a constant rate Brownian motion model and a stable model using StableTraits software version 1.5 (*Elliot and Mooers, 2014*). Two independent Markov chains were run with 2,000,000 iterations at a thinning rate of 200, resulting in 10,000 samples each. Priors on evolutionary rate were set to the default settings as implemented in StableTraits and which prevent rates from approaching zero. The two chains converged after 500,000 iterations as evidenced by a proportional scale reduction factor (PSRF) value approaching 1 (*Brooks and Gelman, 1998*). Therefore, the first 600,000 iterations for each of the two chains were discarded as burn in. The stable model returns a list of median ancestral values for internal nodes along with their 95% credibility interval (*Elliot and Mooers, 2014*). The constant rate Brownian motion model was compared to the stable model using the Bayesian predictive information criterion (BPIC), which is analogous to Akaike's Information Criterion (AIC) in a Maximum Likelihood framework (*Ando and Tsay, 2010*). A vector of median ancestral values of the PC scores was supplied for internal nodes and a phylomorphospace was constructed given the topology of the molecular phylogenetic tree using the 'phytools' package (*Revell, 2012*) in R (*R Core Team, 2017*).

A model-based approach was used to evaluate alternative evolutionary hypotheses: Brownian motion, single-optimum Ornstein-Uhlenbeck (OU), and multi-optimum OU. Prior to evolutionary model comparison, a multivariate extension (*Adams, 2014*) of Blomberg's *K* statistic (*Blomberg et al., 2003*) was computed to estimate the phylogenetic signal in the first three principal components using the 'geomorph' package (*Adams et al., 2017*) in R (*R Core Team, 2017*). The Brownian motion model, which is commonly used in phylogenetic comparative analyses, is defined by the stochastic differential equation (SDE): $dX(t) = \sigma dB(t)$ where $X$ is the trait value, $t$ is time, $dB(t)$ is random white noise, and $\sigma$ is the magnitude of random fluctuations in the evolutionary process. Under Brownian motion all trait changes are independent of previous ones, as well as those on other branches, and trait variance is proportional to time (i.e., branch lengths). Alternatively, the OU process was quantitatively formalized by *Hansen (1997)* to model stabilizing selection as the stochastic differential equation (SDE): $dX(t) = \alpha(\theta - X(t))dt + \sigma dB(t)$ where $\theta$ is the trait optimum and $\alpha$ is the strength of the 'restraining force' acting on a trait around an optimum. Brownian motion is therefore a special case of OU when $\alpha = 0$.

*Hansen (1997)* views phenotypic optima ($\theta$) as peaks in an adaptive landscape, which are a compromise among the many possibly conflicting selective demands acting on a trait at any given time. Multi-optima Hansen models reflect adaptive hypotheses based on observations of extant primate posture and locomotion culled from the literature and quantitatively formalized as alternative arrangements of hypothetical phenotypic optima ($\theta$) following the methodology of *Butler and King (2004)*. Species means represent local optima surrounding a global optimum ($\theta$) which correspond to a selective regime (*Hansen, 1997*). As such, individual species may differ significantly from one another while simultaneously occupying the same global phenotypic optimum, possibly due to other factors such as drift or pleiotropy (*Hansen, 1997*). Therefore, in evolutionary model comparison, the focus is on the number and arrangement of global phenotypic optima ($\theta$) and their surrounding local optima (i.e., species means) rather than on individuals within species. Several recent studies have used modeling methods to test evolutionary hypotheses in paleoanthropology (*Almécija et al., 2015*; *Grabowski and Jungers, 2017*; *Fernández et al., 2018*).

## Evolutionary hypotheses

The evolutionary hypotheses for the link between foot proportions and behavior are informed by observations of locomotor behavior in the wild reported in the literature. The first evolutionary hypothesis is a Brownian motion model (H$_1$). The second evolutionary hypothesis is the first Hansen

model and it reflects a single global phenotypic optimum (H$_2$). Support for these hypotheses would suggest that there are no major adaptive differences in foot proportions between anthropoid primate groups. Subsequent hypotheses represent multi-optima OU models of increasing complexity (i.e., number of phenotypic optima). The second Hansen model has three selective regimes associated with advanced bipedalism (*Homo*), mostly terrestrial quadrupedal locomotion (*Pan*, *Gorilla*, *Papio*, *Theropithecus*, *Erythrocebus*, *Chlorocebus aethiops*) and mostly arboreal quadrupedal locomotion in all other taxa (H$_3$). African apes are competent climbers and possess foot adaptations related to this form of locomotion (*DeSilva, 2009*). However, observations from the wild show them to be highly terrestrial. *Papio*, *Theropithecus*, *Erythrocebus, and Chlorocebus aethiops* are the most terrestrial among the cercopithecoids. Previous studies suggest there may be effects of arboreality and terrestriality on foot proportions (*Schultz, 1963a*; *Jolly, 1967*).

The third Hansen model is an elaboration of the previous one that splits the terrestrial regime into two: terrestrial heel-strike plantigrady in *Pan*, and *Gorilla*, and terrestrial semiplantigrady in *Papio*, *Theropithecus*, *Erythrocebus, and Chlorocebus aethiops* (H$_4$). Semiplantigrady is defined as any habitual foot posture in which some, but not all, tarsals are in contact with the substrate during stance phase. Heel-strike plantigrady is defined as a foot posture in which the tarsals, principally the calcaneus and its proximal tuberosity, contact the substrate at the beginning of stance phase. Several studies have suggested that African ape tarsal morphology reflects their heel-strike plantigrade foot posture (*Gebo, 1992*; *Gebo, 1996*). Previous work on mammalian foot proportions implies that foot proportions may reflect foot posture (*Taylor, 1976*).

The fourth Hansen model separates the arboreal taxa from the previous model into mostly arboreal taxa that engage in little climbing and hindlimb-assisted suspension versus mostly arboreal taxa that engage in climbing and hindlimb-assisted suspension more frequently (i.e., *Pongo abelii*, *Pongo pygmaeus*, *Ateles*, *Alouatta*, *Lagothrix*, H$_5$). Numerous studies have shown that suspensory locomotion is correlated with limb and joint morpology (*Fleagle et al., 1981*; *Gebo, 1996*). The majority of these studies have been focused on the upper limb and shoulder (*Fleagle et al., 1981*; *Gebo, 1996*; *Hunt, 1996*; *Young et al., 2015*). The fifth Hansen model further separates mostly arboreal taxa into those that frequently engage in active climbing (i.e., *Hylobates*, *Symphalangus*, *Alouatta*) from the more hindlimb-assisted suspensory taxa such as *Pongo* and *Ateles* (H$_6$). This model attempts to distinguish between possible effects of hindlimb-assisted suspension versus active climbing (*Sarmiento, 1983*; *Langdon, 1985*; *Harrison, 1986*). Standard model selection criteria (AICc) were used to evaluate alternatives and to choose the model that best fit the data. Alternative adaptive hypotheses were evaluated using the 'OUCH' package (*Butler and King, 2004*) in R (*R Core Team, 2017*).

An additional method was used to identify phenotypic optima without a priori information using the SURFACE method, which stands for SURFACE Uses Regime Fitting with Akaike Information Criterion to model Convergent Evolution (*Ingram and Mahler, 2013*). The purpose of SURFACE is to estimate the macroevolutionary adaptive landscape (i.e., the number and arrangement of phenotypic optima) using only a data set and a phylogeny. SURFACE uses a stepwise AIC algorithm to fit a series of Hansen models in two phases: a forward phase in which selective regimes are added, and a backward phase, in which selective regimes are collapsed. The original intent of the SURFACE method was to test for convergence in a clade while minimizing potential biases in the identification of hypothetically convergent ecomorphs a priori (*Ingram and Mahler, 2013*). These analyses were carried out using the 'SURFACE' package (*Ingram and Mahler, 2013*) in R (*R Core Team, 2017*). Comparison of the model identified by SURFACE (H$_7$) with the a priori models was conducted using the 'OUCH' package (*Butler and King, 2004*) in R (*R Core Team, 2017*). Therefore, a total of seven evolutionary hypotheses describing the evolution of the anthropoid foot were tested. Model comparison offers a powerful method for testing evolutionary hypotheses, but several researchers have noted the importance of conducting simulations to evaluate statistical power in model selection (*Boettiger et al., 2012*; *Cooper et al., 2016*).

Simulations were conducted using a Monte Carlo method in order to evaluate statistical power in model selection closely following the approach outlined in Boettiger and colleagues (*Boettiger et al., 2012*). The purpose of this approach is to determine whether alternative evolutionary models can be distinguished from one another given the data set and phylogeny, and if so, which of the models is best (*Boettiger et al., 2012*; *Lst and Ané, 2014*). It is therefore an alternative to other model selection criteria such as Akaike's Information Criterion (AIC). First, parameters (e.g., log-likelihood) were estimated from the data (i.e., the first three principal components) under

models A and B (e.g, Brownian motion versus single optimum Ornstein-Uhlenbeck). Second, 1000 data sets were simulated under the estimated parameters for each of the two models. Third, models A and B were both re-fit to each of the two 1000 simulated data sets, producing four sets of 1000 log-likelihoods. Finally, the likelihood ratio statistic, which is defined as $\delta = -2(\log L_0 - \log L_1)$, where $\log L_0$ is the log-likelihood of model A and $\log L_1$ is the log-likelihood of model B, was calculated, resulting in two distributions of 1000 values for the likelihood ratio statistic under both models. The difference between the distributions reflects statistical power and the proximity of the empirical likelihood ratio statistic to the distributions indicates which of the two models is best. The following simulations were conducted to compare models of increasing complexity: Brownian motion vs. OU1, OU2 vs. OU3, OU3 vs. OU4, and OU4 vs. OU5. These simulations were conducted using R (*R Core Team, 2017*).

## Acknowledgments

I thank Y Assefa, D Abebaw, the Ethiopian Authority for Research and Conservation of Cultural Heritage (ARCCH), T White, G Suwa, O Lovejoy, and B Asfaw for facilitating access to the fossil specimens of *Ardipithecus ramidus* used in this study. E Westwig (American Museum of Natural History), D Lunde (United States National Museum of National History), and L Jellema (Cleveland Museum of Natural History), JChupasko (Harvard Museum of Comparative Zoology), curatorial staff at the Field Museum, T White (Human Evolution Research Center at the University of California, Berkeley), H Taboada (Center for the Study of Human Origins), C Conroy (Museum of Vertebrate Zoology at Berkeley), W Wendelen and E Gilissen (Royal Museum for Central Africa), M Black and N Johnson (Pheobe A Hearst Museum of Anthropology at the University of California, Berkeley) provided access to museum specimens. S Williams, T Harrison, D Gebo, J DeSilva, and B Wood provided comments that improved this manuscript. C Rolian and two anonymous reviewers provided feedback that improved this manuscript.

## Additional information

### Funding

| Funder | Grant reference number | Author |
|---|---|---|
| Wenner-Gren Foundation | Dissertation Fieldwork Grant | Thomas Cody Prang |

The funders had no role in study design, data collection and interpretation, or the decision to submit the work for publication.

### Author contributions

Thomas Cody Prang, Conceptualization, Formal analysis, Funding acquisition, Investigation, Visualization, Methodology, Writing—original draft

### Author ORCIDs

Thomas Cody Prang (iD) http://orcid.org/0000-0003-3032-8309

### Decision letter and Author response

Decision letter https://doi.org/10.7554/eLife.44433.018
Author response https://doi.org/10.7554/eLife.44433.019

## Additional files

### Supplementary files

• Supplementary file 1. Phylogenetic generalized least squares (PGLS) regression analysis.
DOI: https://doi.org/10.7554/eLife.44433.012
• Supplementary file 2. Performance of alternative models for the evolution of the anthropoid foot.

DOI: https://doi.org/10.7554/eLife.44433.013

• Transparent reporting form

DOI: https://doi.org/10.7554/eLife.44433.014

## Data availability

Data and R scripts have been deposited in Dryad Digital Repository (doi:10.5061/dryad.d112p8r)

The following dataset was generated:

| Author(s) | Year | Dataset title | Dataset URL | Database and Identifier |
|---|---|---|---|---|
| Thomas Cody Prang | 2018 | Data from: The African ape-like foot of Ardipithecus ramidus and its implications for the origin of bipedalism | http://dx.doi.org/10.5061/dryad.d112p8r | Dryad Digital Repository, 10.5061/dryad.d112p8r |

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
