## [Decision Letter]

Thank you for submitting your article "The African ape-like foot of *Ardipithecus ramidus* and its implications for the origin of bipedalism" for consideration by *eLife*. Your article has been reviewed by three peer reviewers, one of whom is a member of our Board of Reviewing Editors, and the evaluation has been overseen by Diethard Tautz as the Senior Editor. The following individual involved in review of your submission has agreed to reveal their identity: Campbell Rolian (Reviewer #3).

The reviewers have discussed the reviews with one another and the Reviewing Editor has drafted this decision to help you prepare a revised submission.

Summary:

*Ardipithecus ramidus* plays a pivotal role in the understanding of homonin evolution and the origin of bipedal locomotion, thanks to the exceptional preservation of its postcranial skeleton. Using the morphometric methods with six variables relating to the foot, Prang tests explicit competing hypotheses of the Homo-Pan LCA and demonstrates that: (1) anthropoid foot proportions represented in *Ardipithecus* are consistent with adaptive optima associated with generally-accepted functional types, (2) *Ardipithecus* shares a number of traits that together closely link it to African apes, and (3) the most parsimonious evolutionary reconstruction of the LCA is African ape-like in proportions, implying that humans and chimps evolved from a semi-terrestrial quadruped. The logic in this manuscript is acceptable but the key conclusion needs more collaboration, as it might raise more debates on the transition close the Homo-Pan LCA node.

Essential revisions:

1) The author used only 6 variables to establish the morphospace of homonin foot. Are these six variables enough to reflect the foot morphospace? Shall the addition of any other variable impact the morphometric affinity between *Ar. ramidus* and Pan plus Gorrilla? The LCA estimation methods should be described in a few more details. Specifically, these methods are known to encounter estimation problems when the multivariate dataset are strongly correlated. It would be good to explain why the PCs are used in lieu of the original 6 GM-corrected variables, and to remind the reader throughout that the input vales are PCs, not original data.

2) A paragraph or a few sentences to remind readers that this is only one small part of the morphology of the LCA, which, although it does support the hypothesis that it may have evolved from plantigrade quadruped resembling African apes, other parts of its morphology, e.g., forelimb, axial skeleton, may show other features that align it more closely to other taxa/morphologies, and on the whole the LCA may still look significantly different from an African ape (much as some colleagues like the ape-like model of LCA, this would be the more cautious approach that would allow for as yet unknown hybrid/mosaic features to emerge from future similar work with other parts of the skeleton).

3) Lovejoy and colleagues have shown their evidences to suggest the 'above branch plantigrady' or 'arboreal multigrady' for *Ar. ramidus*. It looks like the author needs more discussions on their evidences or arguments.

4) The Hansen models could be better described, both in the Results and in the Materials and methods, in particular whether they are hierarchical models that go from simplest to most complex, where complexity is simply the number of optima, or whether each locomotor mode is treated separately and in turn, then each combination of the modes until the full optimum model is achieved.

---

## [Author Response]

Essential revisions:1) The author used only 6 variables to establish the morphospace of homonin foot. Are these six variables enough to reflect the foot morphospace? Shall the addition of any other variable impact the morphometric affinity between Ar. ramudus and Pan plus Gorrilla? The LCA estimation methods should be described in a few more details. Specifically, these methods are known to encounter estimation problems when the multivariate dataset are strongly correlated. It would be good to explain why the PCs are used in lieu of the original 6 GM-corrected variables, and to remind the reader throughout that the input vales are PCs, not original data.

Thank you for asking this question. I conducted a multivariate analysis on 40 geometric mean-standardized variables (first metatarsal distances, fifth metatarsal length, fourth proximal phalanx distances, talus distances, cuboid distances). I collected these data on casts of the ARA-VP-6/500 *Ardipithecus ramidus* foot fossils and they were confirmed on the original fossils at the National Museum of Ethiopia. The *Ar. ramidus* fossil still clusters with the African apes to the exclusion of all other anthropoids. For this study I used fewer variables because it is specifically focused on intrinsic foot (length) proportions and my hypotheses linking morphology and behavior are rooted in the functional morphology of these proportions. The evolutionary modeling and ancestral state estimation analyses were carried out on PC scores to avoid analytical problems associated with correlated variables and to increase statistical power. I added this text: “All evolutionary modeling and ancestral state estimation analyses use the first three principal components (PCs) derived from the PCA. The PC scores were used instead of the original data to avoid analytical problems surrounding correlations among variables (Clavel et al., 2015) and to maximize statistical power (Boettiger et al., 2012).”

2) A paragraph or a few sentences to remind readers that this is only one small part of the morphology of the LCA, which, although it does support the hypothesis that it may have evolved from plantigrade quadruped resembling African apes, other parts of its morphology, e.g., forelimb, axial skeleton, may show other features that align it more closely to other taxa/morphologies, and on the whole the LCA may still look significantly different from an African ape (much as some colleagues like the ape-like model of LCA, this would be the more cautious approach that would allow for as yet unknown hybrid/mosaic features to emerge from future similar work with other parts of the skeleton).

Thank you for raising this point. I added this text: “The *Ar. ramidus* ARA-VP-6/500 partial skeleton is remarkable for its preservation of multiple areas of anatomy (Lovejoy et al., 2009a; White et al., 2009; White et al., 2015). This study provides evidence that intrinsic foot proportions reflect locomotor diversity among anthropoid primates, but it will be important to consider other regions in future comparative studies. The hypothesis that hominins evolved from a semi-terrestrial quadrupedal ancestor could be tested with detailed quantitative analyses of other aspects of *Ar. ramidus* postcranial morphology.”

3) Lovejoy and colleagues have shown their evidences to suggest the 'above branch plantigrady' or 'arboreal multigrady' for Ar. ramidus. It looks like the author needs more discussions on their evidences or arguments.

Thank you. I added this text: “If *Ar. ramidus* and the *Homo-Pan* LCA were adapted to more generalized quadrupedalism and climbing as originally suggested (Lovejoy et al., 2009a; White et al., 2009) or “arboreal multigrady” as later revised (White et al., 2015), then the *Ar. ramidus* foot skeleton should not have been placed in the same regime as the African apes (Figure 3).” Different forms of arboreal quadrupedalism were detected via the evolutionary modeling analyses. For example, *Alouatta* and *Lagothrix* taxa were assigned to their own selective regime among anthropoids. These taxa are arboreal quadrupeds, but they engage in more frequent climbing and suspension than other anthropoids. More generalized quadrupeds were placed into a different regime and include taxa such as *Cercopithecus, Miopithecus*, and *Cebus* (among others). I expected *Ar. ramidus* to be more similar to these kinds of arboreal quadrupeds than to the African apes given what was originally suggested. However, my analyses show that not to be the case.

4) The Hansen models could be better described, both in the Results and in the Materials and methods, in particular whether they are hierarchical models that go from simplest to most complex, where complexity is simply the number of optima, or whether each locomotor mode is treated separately and in turn, then each combination of the modes until the full optimum model is achieved.

Thank you for this suggestion. I added some text in the Materials and methods and throughout the manuscript to make these suggested changes. Here are some examples of the added text: “Subsequent hypotheses represent multi-optima OU models of increasing complexity (i.e., number of phenotypic optima).” and “The evolutionary models differ in increasing complexity where each model includes additional phenotypic optima in a hierarchical manner.”

**Author response image 1. respfig1:** PCA on 40 geometric mean standardized variables (MT1, MT5, PP4, talus, cuboid).